# Universality of the close packing properties and markers of isotropic-to-tetratic crossover in quasi-one-dimensional superdisk fluid

**Sakineh Mizani[1], Martin Oettel[1], Péter Gurin[2]⋆ and Szabolcs Varga[2]**

**1** Institute for Applied Physics, University of Tübingen,
Auf der Morgenstelle 10, 72076 Tübingen, Germany
**2** Physics Department, Centre for Natural Sciences, University of Pannonia,
PO Box 158, Veszprém, H-8201 Hungary

⋆ gurin.peter@mk.uni-pannon.hu

## Abstract

We study equilibrium states and ordering regimes of a quasi-one-dimensional system of hard superdisks (anisotropic particles interpolating between disks and squares) where the centers of the particles are constrained to move on a line. A continuous change from a quasi-isotropic to a tetratic regime is found upon increasing the density. Somewhat unexpected, for isobaric states, systems with larger and more anisotropic particles in the tetratic regime are denser than systems with smaller and less anisotropic particles in a quasi-isotropic regime. Close packing behaviour is characterised by exponents describing the behaviour of the pressure, the angular fluctuations and the angular correlation length. We obtain two universal, shape-independent relations between them.

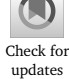

# 1 Introduction

Particle shape and spatial confinement have a fundamental influence on the structural and phase behavior of colloidal suspensions [1, 2] which is particularly intriguing in three dimensions. For instance, rod-like particles such as hard cylinders can undergo a first-order phase transition from an orientationally disordered isotropic phase to an ordered nematic phase as the density increases [3]. Onsager's pioneering work demonstrated that the isotropic-nematic transition can occur even at vanishingly small packing fractions for infinitely long hard rods [4]. Moreover, simulations revealed that the particle shape, rather than just the aspect ratio, is a critical factor in stabilization of different phases. For instance, elongated rod-like cylindrical particles can stabilize smectic A phases, while more compact ellipsoidal particles do not [5–7]. In two-dimensional confinement with a variable slit width, it was shown that a cylindrical shape increases the propensity for nematic ordering compared to an ellipsoid shape [8]. These findings illustrate the subtle interplay between orientational and packing entropies arising from the particle geometry [9].

Spatially confining geometries like slits, cylindrical pores, and rough surfaces further enrich the phase behavior by inducing inhomogeneous density profiles, anchoring and biaxial ordering in the vicinity of confining surfaces [10–18]. In the case of strictly two-dimensional (2D) confinement the freezing is not yet fully understood, even for systems with only excluded volume interactions [19–23]. A well-known example is the freezing of 2D hard disks, which has long been debated whether it follows the Kosterlitz-Thouless-Halperin-Nelson-Young (KTHNY) mechanism involving two continuous phase transitions [24–28]. Simulations suggest that the freezing occurs in two steps, with the first step being a first-order liquid-hexatic phase transition [22]. The situation becomes more intricate when considering anisotropic particles like hard squares, rectangles, and pentagons [29–32]. The competition between packing and orientational entropy further enriches phase behavior with the stabilization of e.g. tetratic, rhombatic and nematic phases [23, 33]. Simulation studies of two-dimensional hard squares showed that at high densities they assemble into a square solid phase, formed in two steps through an intermediate tetratic phase having quasi-long range four-fold orientational and bond orders [29]. For hard pentagons, simulations have uncovered a series of structural transitions, including the formation of a hexagonal rotator crystal phase, followed by the emergence of a striped phase due to geometrical frustration [34]. Experiments on colloidal systems with square and pentagon shapes pointed out that the ordering transitions do not always align with simulation predictions [35, 36]. Li et al. [37] demonstrated that tetratically shaped molecules confined on a spherical surface can develop tetratic orientational order at high molecular density, accompanied by eight disclinations arranged in an anticube configuration. Similarly, Walsh and Menon [38] observed a progression of phases in vibrated hard squares, including a phase with tetratic orientational order, short-range translational correlations, and slowed rotational dynamics.

Here we study quasi-one-dimensional (q1D) systems, bridging the gap between one- and two-dimensional systems. These systems, while seemingly simple, can exhibit unusual phase behavior due to the strong impact of particle shape and the constraints of reduced dimensionality [39–43]. In q1D systems, particle centers are confined to very narrow channels, but retain orientational degrees of freedom. This unique confinement leads again to a rich interplay between translational and rotational degrees of freedom. Systems of hard rectangles confined to move on a straight line exhibit a diverging orientational correlation length, while hard ellipses do not [43]. The introduction of some additional transversal positional freedom as e.g. for hard disks confined between hard parallel walls can lead to the emergence of glassy behavior, jamming, and fragile-to-strong fluid crossovers, even without genuine phase transitions [44–55]. The behaviour of confined hard anisotropic particles such as squares and rectangles is even more complicated [56, 57].

The study of q1D systems is particularly interesting from a theoretical standpoint due to the availability of exact solutions. Starting from the equation of state [58] and correlation functions [59] in the Tonks gas of one-dimensional hard rods, analytical results are also available for wider classes of q1D fluids with attractive interactions [55, 60–63], and even the pair potential can be obtained from pair-distribution functions [64]. In contrast to the more complex phase behavior observed in 2D and 3D systems of anisotropic particles, the nature of phase transitions and the crossovers between different regimes in one-dimensional (1D) systems is generally less controversial [65, 66]. Even though seemingly simple 1D hard body models cannot show genuine thermodynamic phase transitions, they can still exhibit unusual behavior at high pressures [67].

Tuning the shape between sphere and cube (superballs) and between disk and square (superdisk) showed that the varying shape allows for the formation of various solid phases such as plastic and rhombohedral crystals in three dimensions [68] and the so-called $\Lambda_1$ and $\Lambda_2$ phases in two dimensions [69], highlighting the importance of particle geometry in self-assembly processes. However, to the best of our knowledge, there are no studies that focus on the effect of continuously tuning the shape on the ordering behaviour of q1D systems. In addition to this, glassy and jamming behaviors observed in q1D hard disk fluids [44, 49, 70] are not explored in any q1D anisotropic hard body fluids.

The aim of our work is to investigate the role of particle shape in q1D hard-body systems, focusing on the change from continuous angular symmetry to discrete 4-fold symmetry. By deforming circular disk particles into square shapes, we seek to elucidate the structural, orientational, and thermodynamic signatures near close packing where particles become highly ordered. With the help of an exact theoretical transfer operator approach and complementary simulations, we present a detailed analysis of the order parameter, the equation of state, and the orientational correlation for hard superdisk particles confined to move along a line. Central results are the divergent behaviour of pressure and the disappearance of angular fluctuations upon reaching close packing, the peculiar transition behavior from quasi-isotropic to tetratic states and the appearance of certain shape-independent scaling relations.

## 2 Hard superdisk model

We investigate a quasi-one-dimensional (q1D) system comprising $N$ freely rotating hard superdisk particles with centers confined to a one-dimensional (1D) line of length $L$, i.e. the particles have a positional freedom along the $x$ axis and an orientational one ($\varphi$) in the $x$–$y$ plane as shown in Fig. 1. The border of the hard superdisk is described by

$$h := |x|^n + |y|^n - a^n = 0 \,, \tag{1}$$

if the particle's center is located in the origin of the two-dimensional $x$–$y$ plane and $\varphi = 0$. With the deformation parameter $n$ varying between 2 and $\infty$, the particle shape continuously varies from a circular disk to a square. Note that $a$ is half of the side length of the superdisk, which corresponds to the radius of the disk if $n = 2$.

We define by $\sigma$ the (closest) contact distance of two adjacent particles. For two superdisks with orientation angles $\varphi_1$ and $\varphi_2$ at distance $\sigma$, their borders are described by

$$h_1 := |x\cos(\varphi_1) + y\sin(\varphi_1)|^n + |-x\sin(\varphi_1) + y\cos(\varphi_1)|^n - a^n = 0 \,, \tag{2a}$$

$$h_2 := |(x-\sigma)\cos(\varphi_2) + y\sin(\varphi_2)|^n + |-(x-\sigma)\sin(\varphi_2) + y\cos(\varphi_2)|^n - a^n = 0 \,. \tag{2b}$$

Note that $h_1$ describes the border of particle 1, while $h_2$ is for particle 2 (see Fig. 1). When the two particles are in contact, the gradient of $h_1$ and $h_2$ must have opposite direction, i.e.

$$\nabla h_1 = -\mu \nabla h_2 \,, \tag{3}$$

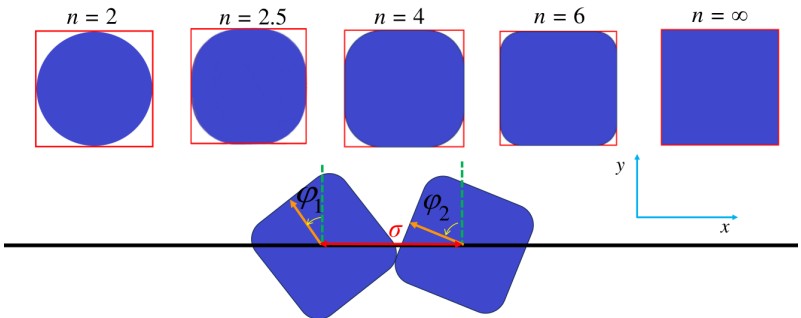

Figure 1: Schematic representation of hard superdisks with varying deformation parameter ($n$). The upper panel demonstrates how the superdisk transforms into the circumscribing square as $n$ increases, while the lower one shows the contact distance between two superdisks having orientation angles $\varphi_1$ and $\varphi_2$, which are measured from the $y$ axis. Superdisks are constrained to move on a straight line ($x$ axis), but are allowed to rotate freely in the $x$–$y$ plane.

where $\mu$ is a positive constant. Eqs. (2) and (3) together provide 4 equations for the unknown $x, y, \sigma$ and $\mu$, where the angles of the neighboring particles and the deformation parameter ($n$) are the input and from which $\sigma$ can be determined numerically. The allowed configurations for particles $(1) = (x_1, \varphi_1)$, $(2) = (x_2, \varphi_2), \dots$ and $(N) = (x_N, \varphi_N)$ satisfy the condition $|x_{i+1} - x_i| \geq \sigma(\varphi_i, \varphi_{i+1})$. For both $\varphi_1$ and $\varphi_2$ close to zero, the contact distance can be approximated by the following analytical expression,

$$\sigma(\varphi_1, \varphi_2) = 2a\left(1 + \frac{\varphi_+^2}{2} - \frac{\varphi_-^2}{2} - \frac{|\varphi_+|^n}{n} + \frac{(n-1)|\varphi_-|^{n/(n-1)}}{n}\right), \tag{4}$$

where $\varphi_+ = (\varphi_1 + \varphi_2)/2$ and $\varphi_- = (\varphi_1 - \varphi_2)/2$. Obviously, the contact distance is equal to the diameter of the disk if $n = 2$, i.e. $\sigma = d = 2a$. In the square-limit ($n \to \infty$), Eq. (4) simplifies to $\sigma = 2a\left(1 + \frac{\varphi_+^2}{2} - \frac{\varphi_-^2}{2} + |\varphi_-|\right)$. We will use Eq. (4) in the analysis of the results. Moreover, in the $P \to \infty$ limit, where all the particles are almost perpendicular to the $x$ axis, the two leading terms are:

$$\sigma(\varphi_-) = d\left(1 + \frac{n-1}{n}\varphi_-^{n/(n-1)}\right). \tag{5}$$

In the following, the side length of the superdisk ($d = 2a$) is the unit of length and is set to 1.

## 3 Theory and simulation method

To study the equilibrium properties of q1D superdisk system, we employ two complementary approaches: a) the transfer operator method [71,72] and b) Monte Carlo simulations [73]. These methods can be used to determine the equation of state, the orientation distribution function, the tetratic order parameter, the angular fluctuations and the orientation correlation length.

The transfer operator method (TOM) is an analytical approach, which is particularly effective for systems with first-neighbor interactions, using the one-dimensional pressure ($P$) as an input parameter. The core of the TOM involves solving the following eigenvalue problem:

$$\int_0^{\pi/2} d\varphi_2 K(\varphi_1, \varphi_2)\psi_i(\varphi_2) = \lambda_i \psi_i(\varphi_1), \tag{6}$$

where $K\left(\varphi_1, \varphi_2\right) = k_B T \frac{\exp(-P\sigma(\varphi_1,\varphi_2)/k_B T)}{P}$ is the basic kernel, $k_B$ is the Boltzmann constant, $T$ is the temperature, and $\sigma\left(\varphi_1, \varphi_2\right)$ is the contact distance between two superdisks. The solution of Eq. (6) are the eigenvalues ($\lambda_i$) and the corresponding eigenfunctions ($\psi_i(\varphi_1)$). We normalize the eigenfunctions as follows: $\int_0^{\pi/2} d\varphi\, \psi_i^2(\varphi) = 1$. Note that the angle is restricted to be between 0 and $\pi/2$ due to the fourfold symmetry of the particle shape. Keeping the order of eigenvalues as $\lambda_0 > \lambda_1 > \lambda_2 > \ldots$, we can determine key thermodynamic and structural properties using the following equations,

$$G = -N k_B T \ln(\lambda_0/\Lambda_{\mathrm{dB}}), \tag{7}$$

$$\frac{1}{\rho} = \frac{dG/N}{dP}, \tag{8}$$

$$f(\varphi) = \psi_0^2(\varphi), \tag{9}$$

$$S = 2\int_0^{\pi/4} d\varphi\, \cos(4\varphi)f(\varphi), \tag{10}$$

and

$$\left\langle \varphi^2 \right\rangle = 2\int_0^{\pi/4} d\varphi\, \varphi^2 f(\varphi). \tag{11}$$

In these equations $G$ is the Gibbs free energy, $\Lambda_{\mathrm{dB}}$ is the de Broglie thermal wavelength, $\rho = N/L$ is the one-dimensional number density, $f(\varphi)$ is the orientational distribution function, $S$ is the tetratic order parameter and $\left\langle \varphi^2 \right\rangle$ describes the angular fluctuation. Using the simplified formula for the contact distance near close packing, Eq. (5), the pressure dependence of the density can be calculated analytically from the eigenvalue equation resulting in

$$\rho^{-1} = d + (2 - 1/n)/(\beta P). \tag{12}$$

The orientational correlation length ($\xi$) is related to the orientational correlation function between particle $j$ and particle $j+i$ as follows

$$g_4(i) = \left\langle \cos\left(4\left(\varphi_j - \varphi_{j+i}\right)\right)\right\rangle - S^2 \sim \exp(-i/\xi). \tag{13}$$

Using the TOM the orientational correlation length can be obtained with the help of two largest eigenvalues [74]

$$1/\xi = \ln\left(\lambda_0/\lambda_1\right). \tag{14}$$

In the case of hard superdisks, the solution of the eigenvalue problem (Eq. 5) requires numerical methods due to the numerically obtained orientation-dependent contact distance. We use the trapezoidal rule in the numerical integrations, with $d\varphi = \pi/10000$, to ensure accuracy in the vicinity of close packing. The eigenvalues and eigenfunctions are obtained using a successive iteration method, with an initial guess of $\psi_i(\varphi) = \sqrt{2/\pi}$ for all systems.

To support and complement our TOM calculations, we perform MC simulations in both *NPT* (constant particle number, pressure, and temperature) and *NLT* (constant particle number, length, and temperature) ensembles. Our simulations typically involve 2000 particles, though this number is increased for very correlated systems. We use the numerically obtained contact distance (the solution of Eqs. (2) and (3)) to implement the overlap condition between the particles. We initialize the system either from a lattice configuration or from a previously equilibrated state. In *NPT* simulations, the system is allowed to equilibrate to its natural density under the applied pressure. Each simulation consists of $2 \cdot 10^7$ Monte Carlo steps (MCSs). One MCS comprises $N$ attempted single-particle moves and rotations. In *NPT* simulations, an additional length change is attempted at each MCS using the standard acceptance criteria. The maximum displacement for translations, the maximum angle for rotation, and the maximum

distance for the length change are adjusted to maintain optimal acceptance ratios, typically aiming for about 40-50% acceptance for each type of move. We characterize the structural properties of the confined superdisks using several measures. The orientational distribution function ($f$) and the order parameter ($S$) are computed with averaging of $f$ and $S$ over $10^5$ snapshots as the snapshots are taken at every 100 MCSs throughout the simulation. The orientational distribution function is calculated using 100 bins spanning the range of possible orientations (from 0 to $\pi/2$, given the symmetry of the superdisks). The orientational correlation function ($g_4(i)$) is also computed from the snapshots to probe the long-range decay of orientational order. The correlation length is determined with plotting $\ln(g_4(i))$ as a function of $i$, where a linear fitting is made on the linear part of the curve. With the linear fitting, the inverse of the negative slope corresponds to the orientational correlation length (see Eq. (13)).

## 4    Results

The one-dimensional system of hard disks ($n = 2$) does not exhibit positional and orientational order in the entire range of the density. The only singularity in its phase behavior is the pressure divergence at close packing as given by the Tonks-equation [58]

$$P = \frac{k_B T \rho}{1 - \rho \sigma}, \tag{15}$$

where $\sigma$ is the contact distance between two hard disks and $\rho = 1/\sigma$ is the close packing density. Intuition suggests that the equation of state of q1D hard superdisks is perhaps similar to that of hard disks if the contact distance is replaced with the orientationally averaged contact distance as follows

$$P = \frac{k_B T \rho}{1 - \rho \langle \sigma \rangle}, \tag{16}$$

where $\langle \sigma \rangle = \frac{4}{\pi^2} \int_0^{\pi/2} d\varphi_1 \int_0^{\pi/2} d\varphi_2 \, \sigma(\varphi_1, \varphi_2)$. With the same side length, $d = 2a$, the average contact distance between two superdisks as well as the area and the average diameter of a superdisk are monotonically increasing functions of the deformation parameter $n$. In this sense, higher $n$ corresponds to "larger" particles. Especially, a superdisk with $n > 2$ is always larger than a disk. Thus, the pressure of superdisks should be higher than that of disks at a given density. Superficially, this can be seen in Fig. 2 (main graph), where the MC simulation and TOM results are shown together. As the deformation parameter ($n$) is increased, the curves are shifted towards higher pressure, but the shape of the curves seems to be qualitatively the same. However, there are differences which can be eventually linked to differences in the ordering behavior between superdisks differing in $n$. We can see this more clearly if $P_n/P_2$ is plotted as a function of density, where $P_n$ is the pressure of hard superdisks having deformation parameter $n$ and $P_2$ is the Tonks pressure of hard disks (superdisk with $n = 2$). It can be seen in Fig. 3 that the equation of state of superdisks is very different from that of hard disks. This manifests itself in the intermediate peak and the close packing values of $P_n/P_2$ (see inset of Fig. 3). Denoting the close packing value of $P_n/P_2$ as $\alpha$, i.e. $\alpha := \lim_{\rho^* \to 1} P_n/P_2$, where $\rho^* = \rho d$, Eq. (12) implies that $\alpha = 2 - 1/n$. This gives $\alpha = 3/2$ for $n = 2$ and $\alpha = 2$ for $n = \infty$, which agree with the result of Kantor and Kardar obtained for hard ellipses and hard rectangles, respectively [43] and implies that $\alpha$ is discontinuous at $n = 2$, showing a jump from 1 to 1.5, related to the fact that the rotational symmetry of the particle changes abruptly from a continuous SO(2) symmetry to a discrete fourfold $C_4$ symmetry. Interestingly the largest deviation between $n > 2$ and $n = 2$ cases occurs at intermediate densities, where $P_n/P_2$ is maximal. This is shown in the inset of Fig. 3. In the limit $n \to 2$, we obtained numerically that the maximum of $P_n/P_2$ goes to 1.75, i.e. this maximum also exhibits a jump at $n = 2$ due to
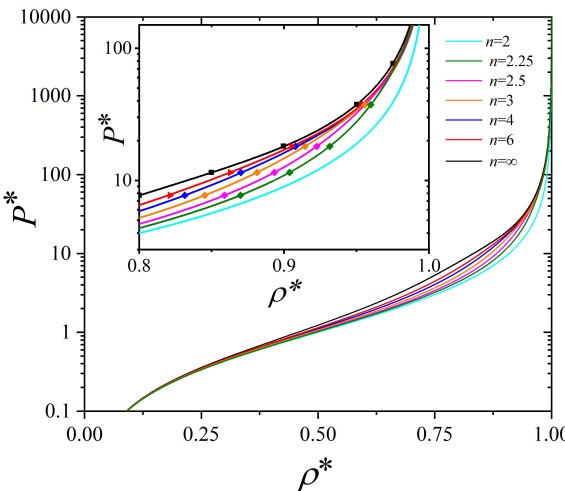

Figure 2: The effect of the deformation parameter ($n$) on the equation of state of q1D hard superdisks. The inset highlights the high-pressure behavior and shows the TOM (lines) and the MC data (symbols). A dimensionless pressure and density are defined by $P^* = Pd/k_B T$ and $\rho^* = \rho d$.

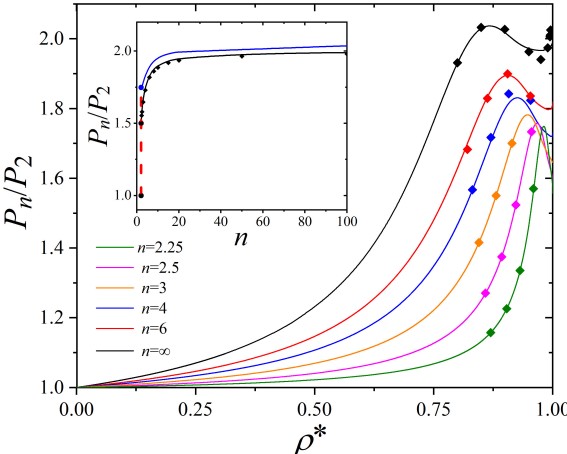

Figure 3: The deviation between the equation of state of hard superdisks and that of hard disks in the $P_n/P_2$ vs. $\rho^*$ plane (lines: TOM, symbols: MC data). For squares, $n = \infty$, the highest pressure simulated is $P^* = 410$, while for all other cases it is only $P^* = 195$. It can be seen that at very high pressures, in the vicinity of close packing density, the simulations are inaccurate. The inset shows the maximum of $P_n/P_2$ (blue curve from TOM) and the high pressure (close packing) limit of $P_n/P_2$ (black diamonds) as a function of $n$. The vertical dashed line in the inset indicates the discontinuities occurring in $P_n/P_2$ at $n = 2$. The analytic form $\alpha = 2 - 1/n$ is shown as a continuous black curve.

the change of symmetry in the shape. The last interesting result in Fig. 3 is that some curves intersect each other and consequently larger particles can have a lower pressure than smaller ones at equal densities, which contradicts Eq. (16). This peculiar behaviour of hard superdisks is seen more clearly in Fig. 4 (main graph), where the equation of states of $n = 2.1$ and $n = 4$ cases are shown together at high densities. The pressure of the smaller particles ($n = 2.1$) is higher than that of larger particles ($n = 4$) between the two densities at which the two equation of state curves intersect. This pressure inversion occurs only in a very narrow density window,

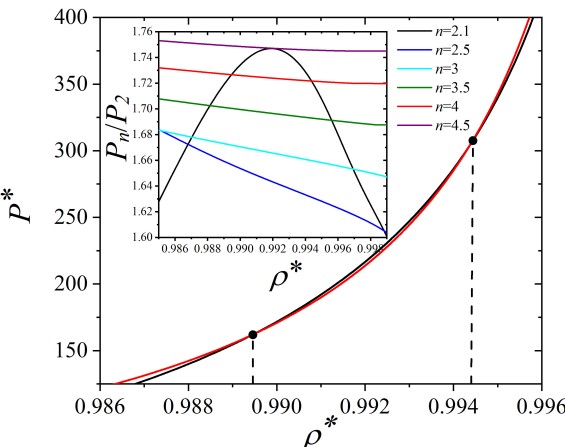

Figure 4: The equation of state in the vicinity of close packing density ($\rho^* = 1$) in $P^*$ vs. $\rho^*$ (main panel) and $P_\mathrm{n}/P_2$ vs. $\rho^*$ (inset) planes. The vertical dashed lines and the filled circles delimit the density region where the smaller superdisks ($n = 2.1$) have higher pressure than the larger ones ($n = 4$).

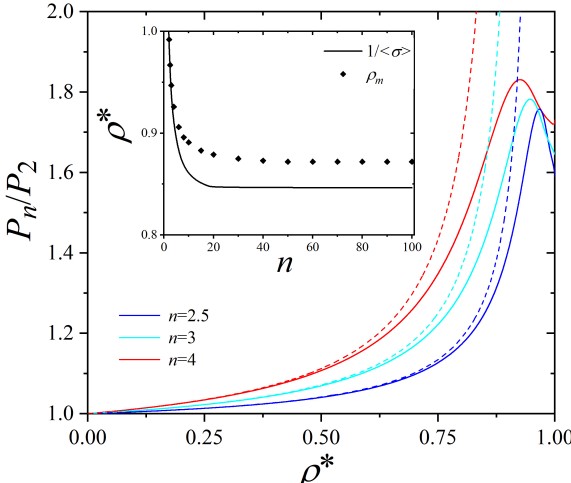

Figure 5: Comparison of the pressure ratios ($P_\mathrm{n}/P_2$) of hard superdisks (continuous curve) and that of hard disks (dashed curve) interacting with the average contact distance of hard superdisks. The inset shows an estimate, $1/\langle\sigma\rangle$ (continuous curve) for a maximal isotropic density as a function of deformation parameter ($n$), while the diamond symbols indicates $\rho_m$, which are the values of $\rho^*$ obtained by TOM where ($P_\mathrm{n}/P_2$) is maximal.

while the systems exhibit "normal" behaviour below first cross point and above the second one. The inset of Fig. 4 further shows the pressure ratio for $n = 2.1$ superdisks compared to $n = 2.5 \dots 4.5$ in the intersection region. We observe that the density window of the pressure inversion shrinks with increasing $n$, and it disappears completely at $n = 4.5$. Thus, pressure inversion occurs only if the difference between the sizes of the smaller and larger particles is not too high.

To understand the mechanism of pressure inversion, the emergence of the $P_n/P_2$ peak needs to be clarified. To do this, we plot $P_n/P_2$ coming from Eq. (16) together with the exact TOM solution of Eq. (6) in Fig. 5. According to Eq. (16), $P_n/P_2$ should diverge at $\rho = 1/\langle\sigma\rangle$. This is however a spurious consequence of assuming an isotropic angular distribution in Eq. (16), which cannot be true at very high densities. If superdisks are ordered fully

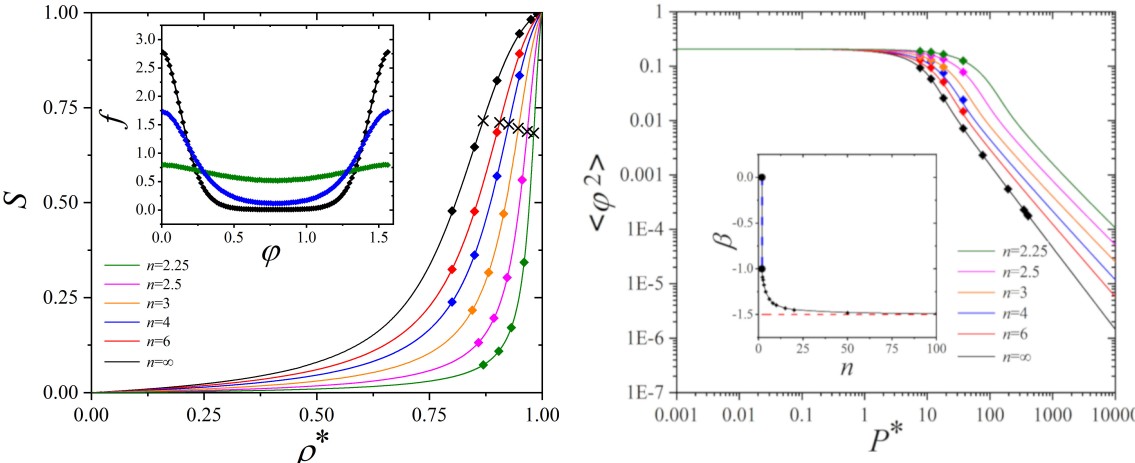

(a) The tetratic order parameter as a function of density for various $n$ (cross symbols indicate the state where $P_n/P_2$ is maximal). Inset: the orientational distribution function as a function of angle at $\rho^* = 0.9$ for $n = 2.25$ (green, quasi-isotropic state), $n = 4$ (blue, transitory state) and $n = \infty$ (black, tetratic state).

(b) The angular fluctuation as a function of pressure (symbols: MC data, lines: TOM). Inset: the corresponding exponent, $\beta$, as a function of $n$. (lines: TOM, symbols: fitted values of $\beta$ from MC data). The vertical dashed line indicates the discontinuity of $\beta$ at $n = 2$, while the horizontal one denotes the limiting value at $n = \infty$.

Figure 6: The effect of deformation parameter ($n$) on the tetratic ordering.

parallel, the close packing density is the same ($1/d$) for all $n$, and $\rho = 1/\langle\sigma\rangle$ can be considered just as a maximal isotropic close packing density. One can see on the inset of Fig. 5 that the density at which $P_n/P_2$ is maximal decreases with increasing $n$ similarly as $1/\langle\sigma\rangle$ does. Therefore, the peak in $P_n/P_2$ can be considered as a marker of a structural change from a weakly ordered quasi-isotropic to an orientationally strongly ordered tetratic regime. The emergence of the maximum in $P_n/P_2$ is due to the competition between orientational and packing entropies. On the left side of the $P_n/P_2$ peak, the orientational entropy is more dominant than the packing entropy, while on the right side the opposite occurs. Since the division of entropy into sums of different terms is somewhat arbitrary, to avoid confusion, we define explicitly the different entropy terms. The translational entropy is given by $S_t = Nk_B(1 - \log\rho\Lambda)$, and the orientational entropy is $S_o = -Nk_B \int d\varphi\, f(\varphi)\log f(\varphi)$. These two terms give the ideal part of the free energy, $F_{id} = -T(S_t + S_o)$, while the whole remaining part is called excess term, i.e. the total (exact) free energy can be written as $F = F_{id} + F_{exc}$. The excess part is due to the interaction, in our case the hard body repulsion, which restrict how the particles can be packed avoiding overlaps. Therefore the related entropy is called packing entropy, $S_p = -F_{exc}/T$. Note that the competition between the orientational and packing entropies gives rise to disorder–order phase transitions in higher dimensions. If the orientation entropy wins over the packing entropy, the phase is isotropic, whereas if the packing entropy wins, the phase is ordered [3, 9]. The reason why Eq. (16) worsens in the vicinity of $P_n/P_2$ peak is that it overestimates (underestimates) the contribution of orientational (packing) entropy. Therefore Eq. (16) can be considered as an equation of state of the perfect isotropic system, which cannot describe orientationally ordered system.

To illustrate the relation between orientational ordering and the decrease occurring in $P_n/P_2$, Fig. 6a shows the tetratic order parameter as a function of density and Fig. 6b the orientation angle fluctuation as function of pressure. At fixed density, tetratic ordering becomes stronger when moving from disks to squares. For the particular density $\rho^* = 0.9$, the inset of Fig. 6a demonstrates that the orientational distribution function is practically isotropic for $n = 2.25$ (green curve), while it is strongly ordered tetratic for $n = \infty$ (black curve).

We can also see in Fig. 6a that the particles are still only moderately ordered ($S < 0.75$) at $\rho = 1/\langle \sigma \rangle$, while the tetratic order parameter is very low ($S < 0.25$) at $\rho = 1/d_{\max}$. Here $d_{\max}$ is the length of the particle's diagonal, thus $1/d_{\max}$ is a characteristic density below which orientation effects are to be expected unsignificant. Note that even weak tetratic ordering has an effect on $P_n/P_2$ resulting in some deviation between Eq. (16) and the exact solution (see Fig. 5). The maximum of $P_n/P_2$ is not a sharp transition point between the isotropic and tetratic regime (as expected), but it divides the regions of weakly and strongly ordered structures. Therefore, the left side of the $P_n/P_2$ peak can be considered as a quasi-isotropic regime, while the right side a tetratic regime. This confirms that the $P_n/P_2$ peak and the corresponding density can be considered as a marker of structural change between quasi-isotropic and tetratic regime. Interestingly as $n$ is changed between 2 and $\infty$, the order parameter varies between 0.68 and 0.71 at the $P_n/P_2$ peak (see Fig. 6), i.e. $S \approx 0.7$ can be considered as a second marker of the crossover.

In the light of above, the pressure inversion shown in Fig. 4 can be explained as follows. The smaller (less anisotropic) superdisks are in a quasi-isotropic regime, while the larger (more anisotropic) particles form a very ordered tetratic regime at the left border of the pressure inversion density. This results in a higher effective contact distance between the smaller particles than the contact distance between the almost parallel larger ones. The right density border of the pressure inversion appears since also the smaller particles order parallel, and the same angular fluctuations result in a higher contact distance between the larger and more anisotropic particles than between the smaller ones. Particles with a high $n$ are always "effectively larger" than particles with a small $n$ (due to angular fluctuations), thus the pressure inversion disappears with increasing size difference (as e.g. superdisks with $n > 4.5$ compared with $n = 2.1$).

We now proceed to understand the role of angular fluctuations, which should vanish as the system becomes perfectly ordered in the tetratic regime ($S \rightarrow 1$). From Fig. 6b one sees that, up to $P^* \approx 2$, $\langle \varphi^2 \rangle$ is constant for all values of $n$ and close to $\pi^2/48 \approx 0.21$ which is the value of the angular fluctuation in the isotropic regime, indicating that the system is in a "quasi-isotropic" regime. After an intermediate transition regime, $\langle \varphi^2 \rangle \propto P^\beta$. Our numerical fit for $\beta$ is presented in the inset of Fig. 6b. It can be seen that $\beta$ is discontinuous with a jump from 0 to $-1$ at $n = 2$ and goes to $-3/2$ with increasing $n$, which is the limiting value of hard squares. The simplest equation, which can describe this change is $\beta = 1/n - 3/2$. This equation perfectly fits the numerically obtained data (inset of Fig. 6b). Note that our results for $\beta$ are consistent with $\beta = -1$ and $-3/2$ values obtained for hard ellipses and rectangles, because $n = 2$ for the ellipse and $n = \infty$ for the rectangle [43]. We observe that the decay of the angular fluctuations becomes steeper with the increasing shape anisotropy (increasing $n$) in the vicinity of close packing.

The pressure dependence of the orientational correlation length is presented in Fig. 7. This quantity increases linearly above $P^* \approx 100$ for all $n$ values in $\log \xi$–$\log P^*$ plane, which means that $\xi$ is proportional to $P^\gamma$. A numerical fit to the linear part of Fig. 7 provides the values of the exponent $\gamma$ (inset of Fig. 7), showing that $\gamma$ starts from 0 and converges to $1/2$ for $n \rightarrow \infty$. The value $\gamma = 0$ obtained for $n = 2$ is reasonable, because the hard disks are orientationally uncorrelated, while the value $\gamma = 1/2$ for $n = \infty$ is identical with the value obtained for hard rectangles [43]. The simple form $\gamma = 1/2 - 1/n$ fits the numerically obtained results very well. Note that $\xi$ increases faster than the asymptotic power law up to $P^* \approx 20$, which supports our finding about the structural change from quasi-isotropic to tetratic order.

The divergence of the orientational correlation length (for $n > 2$) upon pressure increase near close packing can be rationalized by an inspection of single-particle, local angular fluctuations $\delta \varphi_{\mathrm{loc}}$ vs. the global angular fluctuation $\delta \varphi_{\mathrm{glob}} = \sqrt{\langle \varphi^2 \rangle}$ [43]. On the one hand, to obtain the pressure dependence of $\delta \varphi_{\mathrm{loc}}$ near the close packing, we realize that the local angular fluctuations are related to the local positional fluctuations, $\delta x_{\mathrm{loc}}$, via the contact distance,

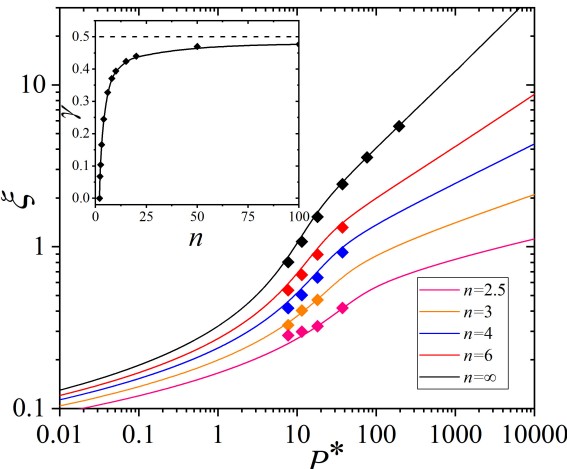

Figure 7: The effect of deformation parameter ($n$) on the orientational correlation length ($\xi$). The main panel shows $\xi$ as a function of $P^*$ (symbols: MC data, lines: TOM). Inset: corresponding exponent $\gamma$ (with $\xi \sim P^\gamma$) as a function of $n$ (symbols: fitted values of $\gamma$, continuous curve: the analytic form $\gamma = 1/2 - 1/n$). The dashed horizontal line corresponds to the limiting value of hard squares ($n = \infty$).

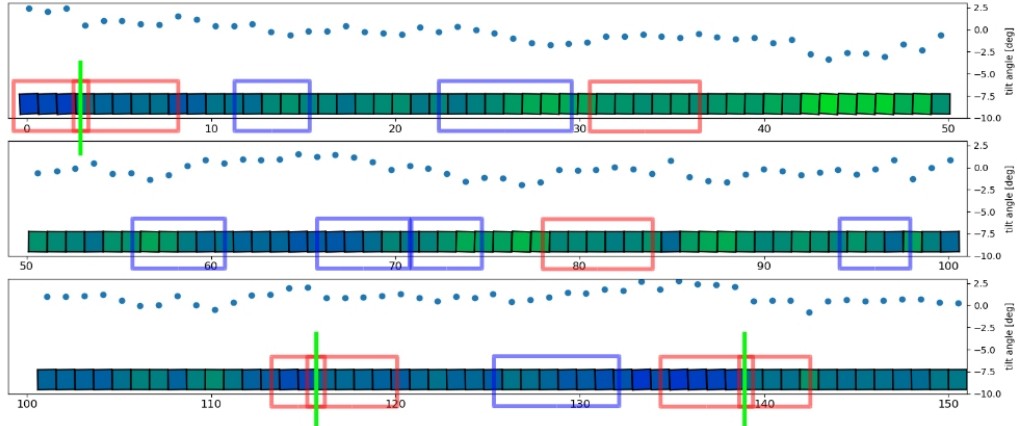

Figure 8: Part of a MC snapshot of q1D hard squares at $\rho^* = 0.99$. For clarity, the tilt (orientation) angle is coded in the colour of the squares and also shown as points above the particle (right y axis). Some domains of type (a) with (approximately) constant tilt angle are marked by red rectangles and a green line shows a defect, separating two such domains. Other domains of type (b) (where the orientation angle increases or decreases) are marked by blue rectangles.

$\delta x_{\text{loc}} \approx \sigma(\delta\varphi_{\text{loc}})$, where $\sigma$ is given by Eq. (5). From this, we have $\delta\varphi_{\text{loc}} \propto (\delta x_{\text{loc}})^{(n-1)/n}$. Moreover, $\delta x_{\text{loc}}$ is nothing else but the average free space between the neighbouring particles, $\delta x_{\text{loc}} = 1/\rho - d \propto 1/P$, see Eq. (12), thus we infer a local angular fluctuation $\propto P^{(1-n)/n}$. On the other hand, the global fluctuation is $\delta\varphi_{\text{glob}} \propto P^{\beta/2} = P^{-3/4+1/(2n)}$. Therefore, the ratio of local to global fluctuation $\delta\varphi_{\text{loc}}/\delta\varphi_{\text{glob}} \propto P^{-1/4+1/(2n)}$ is going to zero towards close packing for $n > 2$. Thus, the system can build such global fluctuations only via a large number (going to infinity at close packing) of correlated local fluctuations [43]. This can be realized in two ways: (a) via domains of parallel particles, possessing the same orientation, and (b) "rotating" domains in which the particle orientation angle is either increasing or decreasing. For illustration, we show part of a MC simulation snapshot of squares in Fig. 8 ($n = \infty$, $\rho^* = 0.99$, $P^* \sim 195$)

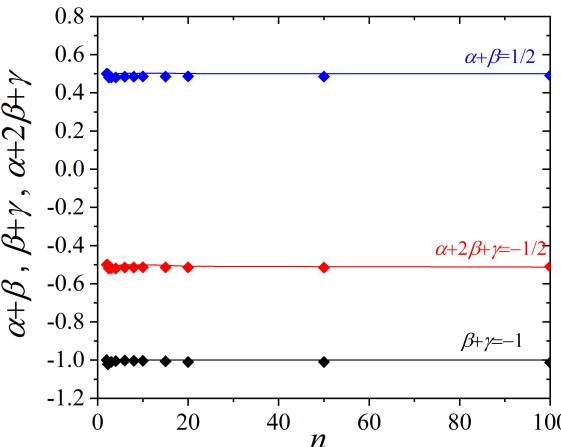

Figure 9: The combinations $\alpha+\beta$, $\alpha+2\beta+\gamma$ and $\beta+\gamma$ as a function of $n$. The symbols are fitted values from the TOM results, while the continuous lines show the constants resulting from the exact value $\alpha = 2 - 1/n$ and the assumptions $\beta = 1/n - 3/2$, and $\gamma = 1/2 - 1/n$.

where examples of domains of type (a) are marked with red rectangles and those of type (b) with blue rectangles. Both types of domains, with fixed tilt angle and "rotating" ones, have an extend of a few particles (less than 10), in accordance with the correlation length 5.6 obtained from TOM. The domains of type (a) appear to be orientationally locked.

Finally, we present some combinations of the high pressure quantities in Fig. 9. The analytic formulas for $\alpha$, $\beta$ and $\gamma$ result in constant values for $\alpha+\beta$, $\alpha+2\beta+\gamma$, and $\beta+\gamma$, i.e., these combinations do not depend on the superdisk deformation parameter. We take $\alpha+\beta = 1/2$ and $\beta+\gamma = -1$ as basic scaling relations for the universal behaviour of q1D hard superdisks, while $\alpha + 2\beta + \gamma = -1/2$ can be derived from these two equations. The first equation ($\alpha + \beta = 1/2$) couples the close packing behaviour of $P_n/P_2$ and the decay of angular fluctuations at very high pressures, while the second equation ($\beta + \gamma = -1$) connects the angular fluctuations and correlations. As $\alpha = P_n/P_2$ in the close packing limit ($\rho^* \to 1$) is a measure of the deviation from the q1D hard disks' (Tonks) equation of state, $\alpha+\beta = 1/2$ means that the larger (smaller) deviation in the equation of states result in a weaker (stronger) angular fluctuations. The relation $\beta + \gamma = -1$ means that a faster (slower) decay of the angular fluctuations is always accompanied by longer (shorter) orientational correlation.

## 5 Conclusion

We studied the effect of varying shape on the ordering behavior of quasi-one-dimensional hard superdisks which encompasses an abrupt change from a continuous SO(2) symmetry to a discrete fourfold $C_4$ symmetry at the value $n = 2$ of the deformation parameter. This discontinuity of the rotational symmetry implies that the behaviour of the angular fluctuations and the equation of state are discontinuous at close packing as a function of $n$: $\alpha$ jumps from 1 to 1.5 and $\beta$ jumps from 0 to $-1$ at $n = 2$, where $\alpha := \lim_{\rho^* \to 1} P_n/P_2$ and $\langle \varphi^2 \rangle \sim P^\beta$. The exponent for the orientational correlation length ($\xi \sim P^\gamma$) is not discontinuous since $\gamma$ is 0 at $n = 2$ and the system stays uncorrelated in the limit of vanishing fourfold symmetry. The most remarkable result of our study is that the close packing exponents ($\alpha$, $\beta$ and $\gamma$) fulfill two shape independent relations, namely $\alpha + \beta = 1/2$ and $\beta + \gamma = -1$. This means that the pressure, the orientational distribution and the angular correlation are closely linked. E.g., smaller angular fluctuations give higher pressures at close packing via $P_n/P_2 = 1/2 - \beta$, and

from $\xi\langle\varphi^2\rangle \sim 1/P$ one sees that the decay of $\langle\varphi^2\rangle$ is faster than the divergence of $\xi$. Regarding the shape dependence of the exponents, $\alpha$ and $\gamma$ increase, while $\beta$ decreases with $n$. Therefore, the pressure ratio ($P_n/P_2$) increases as the shape of the particle becomes more anisotropic, i.e. the pressure of more anisotropic particles is higher than that of less anisotropic ones. Hence, q1D hard superdisks are never identical to q1D parallel hard superdisks (hard rods), because effects of angular fluctuations are present even near close packing. The contribution of the angular fluctuations in the pressure is $\Delta P = (1 - 1/n)P_2$, which gives an excess pressure $\Delta P = P_2/2$ and $\Delta P = P_2$ at close packing for $n \to 2$ (disk limit) and $n \to \infty$ (square limit), respectively.

The other important result of our study is that the emerging intermediate peak in $P_n/P_2$ can be considered as a marker of isotropic-tetratic crossover. Analyzing $P_n/P_2(\rho)$ in a q1D hard superellipse system could reveal whether this quantity is also capable of marking the transition between quasi-isotropic and nematic regimes. Moreover, $P_n/P_2$ may constitute a good marker of other structural changes such as the fluid-zigzag structural change happening in single-file hard disk systems [47]. Therefore, the density dependence of $P_n/P_2$ should be examined in other q1D systems without true phase transitions to see the predicting power of $P_n/P_2$ for structural changes.

In perspective, we speculate that the scaling relations $\alpha + \beta = 1/2$ and $\beta + \gamma = -1$ are also valid for superellipses (interpolating between ellipses and rectangles). In this case, the aspect ratio ($k$) and the deformation parameter ($n$) play the key role in the formation of orientationally ordered structure. If $k > 1$, the system has two-fold symmetry and the superellipses form a nematic phase at high pressures. According to Kantor and Kardar [43], $\alpha$, $\beta$ and $\gamma$ do not depend on the aspect ratio, as $\alpha = 1.5$ (2), $\beta = -1$ (-1.5) and $\gamma = 0$ (0.5) for the ellipses (rectangles), which agree with our $\alpha = 2 - 1/n$, $\beta = 1/n - 3/2$ and $\gamma = 1/2 - 1/n$ findings. The check of the generality of $\alpha + \beta = 1/2$ and $\beta + \gamma = -1$ relations for other shapes and confinement is left for future studies.

# Acknowledgments

**Funding information**  S. M. acknowledges financial support from the Alexander von Humboldt Foundation. P. G. and S. V. gratefully acknowledge the financial support of the National Research, Development, and Innovation Office – NKFIH K137720 and the TKP2021-NKTA-21.

**Data and code availability**  All data and codes are available on request from S. M. or S. V. (sakineh.mizani@mnf.uni-tuebingen.de, varga.szabolcs@mk.uni-pannon.hu).

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
