# Peer review of "Universality of the close packing properties and markers of isotropic-to-tetratic crossover in quasi-one-dimensional superdisk fluid"

_SciPost Physics Core, doi:SciPost Phys. Core 8, 008 (2025)_

## Round 1 · Referee Report · Anonymous (Referee 1) · 2024-10-31

Strengths
1) The manuscript provides a thorough exploration of the behavior of a clean family of model systems, which can be seen as limiting cases for more general problems involving particles confined in narrow channels. Hence, I expect that the presented results will be useful as a reference state for future studies.
2) The manuscript is well written and presented, and is easy to follow.
Weaknesses
1) I am not fully convinced by the manuscript's arguments about orientational and translational entropy (see report).
2) Some minor inconsistencies/typos in equations.
Report
The paper is well written, and provides useful theoretical results for a clean family of model systems with interesting behavior. As the model system can be seen as a limiting case for more general problems involving particles confined in narrow channels, I expect that the presented results will be useful as a reference state for future studies. Hence, I think the manuscript should be published. That said, I have a number of questions and comments the authors should address, which I provide in detail below.
Requested changes
1) Eqs. 2a and 2b appear to be missing a factor $y$ in front of the last cosine term.
2) I'm confused about the dimensionality of variables in Eqs. 6-7 (plus the non-numbered inline equations in between). From the definition of K, it appears to have units of length: the exponent is dimensionless, so [K] = energy / pressure = length. This means its eigenvalues have dimension of length too. But then in Eq. 7 $\lambda_0$ is treated as a dimensionless number. My guess is that a thermal wavelength is missing?
3) The initial guess the authors use for $\psi_i$ appears to be inconsistent with their normalization.
4) In the inset of Fig. 5, the authors plot two estimates of densities where one expects the shape of the particle to start playing a dominant role. In the text, the authors argue that the density at which $P_n / P_2$ is maximal decreases similarly to one of these estimates. It would probably be helpful to include the behavior of this density (the one where $P_n / P_2$ is maximized) in the same inset, so that the readers can compare the behavior.
5) There are some deviations between the theoretical and simulation results, especially at high $n$. Can the authors speculate on the reasons for the deviations?
6) The authors associate the peak in $P_n/P_2$ with a competition between orientational and packing entropies. However, I find this explanation a bit unsatisfying in its current form. In particular, the authors say that on the left side of the $P_n / P_2$ peak, the orientational entropy is more dominant than the packing entropy, while on the right side the opposite occurs.
I am not sure I agree with this description. In principle, if we split the total entropy of the system into $S_\mathrm{tot} = S_\mathrm{trans} + S_\mathrm{rot}$, then I would expect the pressure to be associated with the sum of the volume-derivatives of the two entropy terms, so the total pressure could similarly be decomposed into $P_\mathrm{trans}$ and $P_\mathrm{rot}$. At low densities, I would then argue that the effect of translational entropy dominates the behavior of the pressure, since the derivative of the rotational entropy with respect to the volume is close to zero at low density: the rotational freedom really only starts to get restricted (impacing Srot) at high densities. So in the ideal gas limit, the pressure must be dominated by translational entropy, opposite of the authors' argument about the left side of the peak.
Hence, if the authors want to make a claim about the changes in the role of orientational and translational entropy on either side of the peak, it would be good to actually make an estimate of these entropies to base this argument on.
7) The authors state that Eq. 16 overestimates the contribution of orientational entropy. However, Eq. 16 does not take any orientational freedom into account: it considers a Tonks gas of particles with a fixed average size, which can be regarded as a system where the orientations of the particles are frozen. Hence, I would argue that it contains no contribution from orientational entropy at all. Simultaneously, the expression clearly underestimates the value of the packing entropy, since the particle size is overestimated at finite densities.
8) On page 10, the authors write that in the regime where the pressure inversion is observed, there is "a higher effective contact distance between the smaller particles than the contact distance between the almost parallel larger ones." Is this a measurement from simulation, an observation from the theoretical results, or an argument based on the behavior of the pressure?
9) The authors argue that at high pressure, the local fluctuations in angle are small compared to the global fluctuations, and hence the global fluctuations must arise from correlated local fluctuations. Is this essentially equivalent to saying that the main contribution to variation in the angle comes from large-wavelength fluctuations, since small-wavelength ones are strongly suppressed by the hard-core interactions? (This is mostly just a question out of curiosity.)
10) On page 12, the authors write that for a weakly jammed state the density can decrease without increasing the total length of the system. However, in a 1D system, I do not understand how the density could decrease without either increasing the volume (which means increasing the length of the system), or reducing the number of particles (which I do not think the authors consider here).
11) Overall, I am not entirely happy with the line of argument around strongly and weakly jammed states for discussing the behavior of the correlation length. In my mind, the correlation length is a static quantity of the system, independent of the imposed dynamics. In contrast, the weakly jammed states are discussed in terms of dynamics, which should not affect static quantities. For example, the authors argue that the state in Fig. 9b can only change due to collective motion or via progressive melting'' from from the border of the jammed region, and state that the "probability of both processes is low". However, this depends on the chosen dynamics. In Monte Carlo, the dynamics may be slow, because one typically does not include collective moves, and moves in the center of the
jammed'' region will be rejected. However, if the particles are following Newtonian dynamics (as e.g. in a molecular dynamics simulation), collective dynamics would rapidly relax the illustrated state. Despite the changes in relaxation time, the choice of dynamics would not affect the equilibrium correlation length, and hence I would avoid mixing the discussion of dynamical behavior and static correlations.
12) In the introduction, the authors write "Spatially confining geometries ... further enrich the phase behavior by inducing inhomogeneous density profiles, anchoring, and biaxial ordering in the vicinity of the confining surfaces. In this regard, the freezing of two-dimensional systems is not yet fully understood... ". I do not really see how the second sentence is related to the first. True 2D systems, such as the 2D hard disks the authors discuss immediately after this, do not exhibit any of the mentioned features of confining geometries, unless the 2D systems are under explicit confinement as well.
13) In the discussion of 2D phase behavior on page 2, Ref. 33 seems to be about 3D cubes, in contrast to the 2D shapes mentioned in the same sentence.
14) Page 2: "In systems of hard rectangles confined to move on a straight line exhibit a diverging orientational correlation length". The "In" at the start should probably not be there.
15) Page 4: "perpendicular to the x axes". "axes" should be "axis".
Recommendation
Ask for major revision
Author: Péter Gurin on 2024-11-11 [id 4956]
(in reply to Report 1 on 2024-10-31)
We thank the Referee for the detailed and helpful work. Our answers are below.
1. The Referee is right, we correct the manuscript.
2. The Referee is right, we treated the physical dimensions in a sloppy, imprecise way. The reason is that the Gibbs free energy shifts by only an insignificant constant when the thermal de Broglie wavelength is taken into account, and it has no impact on the further results, such as on Eq. 8.
Thank you for the comment, we make or manuscript more precise.
3. Thanks, it was a misprint, we have corrected.
4. Thank you for this helpful suggestion, we have done so. In our opinion the message of this part of the paper is clearer in this way. The figure now clearly shows that the density at which $P_{n} / P_{2}$ is maximal correlates very well with $1 /\langle\sigma\rangle$, i.e. this density can be considered as a marker of a structural change from a weakly ordered quasi-isotropic to an orientationally strongly ordered tetratic phase. In the light of this good correlation we decide to remove $1 / d_{\max }$, which is a worse estimate, and have made further minor changes on the manuscript accordingly.
5. The reason of the deviation is not related with the value of $n$ but the value of the density. In the very vicinity of the close packing density the simulations are more difficult. For squares, $n=\infty$, the highest pressure simulated is $P^*=P d / k_{\mathrm{B}} T =410$, while for all other cases it is only $P^*=195$. It can be seen that at very high pressures, very close to the close packing density, the simulations are inaccurate. We include this information in the revised version of the manuscript. Actually, since we have experienced this for the squares, this is the reason why we have not done simulations at such high pressures for the other cases.
6. The referee is right that the translational entropy dominate the low density behaviour of the system, not the orientational one. However, in the mentioned part of our manuscript we refer to the competition between the orientational entropy and the packing entropy, not the translational entropy. We have not defined explicitly what we mean by the different entropy terms, and this may have led to misunderstanding. As the Referee highlighted this problem as a weakness of our article, here we clarify this issue in detail and add some definitions to the manuscript to avoid the confusion. We focus only on one-dimensional hard body systems.
In general, free energy can be divided into an ideal and an excess part,
However, the decomposition of entropy into a translational part, an orientation part and a packing part is somewhat artificial, since $a_{exc}$ also depends on $f$. In fact, $a_{exc}$ is a function of $\rho$ and a functional of $f$, which also depends on $\rho$. This problem is even more pronounced when we consider the contribution of these entropies to the pressure, as the Referee has done.
Despite these difficulties the ordering behaviour of the system, i.e. the ODF is determined by the fact that in equilibrium $s$ has a maximum, and $s_o$ prefers the orientational disorder, it has a maximum in the isotropic phase, while $s_p$ prefers the complete order, when all the particles are parallel. Therefore the competition of these terms determine the form of $f$. This argument appears in our paper.
We believe that our argument is valid, therefore we do not want to change it, but we are open for further discussion if we misunderstood the Referee's criticism.
However we do not use the term "translational entropy" in the criticized part of our manuscript, but we use this term in a similar context in the Introduction. (This shows that this terminology is not very canonical.) In the new version of the manuscript we everywhere use the "packing vs. orientational entropy" terms. Furthermore, to clarify the terminology used in the manuscript we include the definition of the different entropy terms.
7. In Eq. 16, the contribution of the orientation entropy does not appear because it disappears at the derivation. However, from the point of view of the (orientational) entropy, there is a difference between the original Tonks gas and the system of freely rotaing disks. In the former case, as the Referee write, the orientations of the particles are frozen, and the dominant eigenvalue of the transfer matrix is $\lambda_0=\frac{e^{-\beta P d}}{\beta P}$. While in the later case, due to the integral over the orientational degrees of freedom, $\lambda_0=2\pi \frac{e^{-\beta P d}}{\beta P}$. Taking the logarithm we obtain an extra $k_B \log 2\pi$ part in the entropy per particle, $S/N=-\frac{\partial G/N}{\partial T}$, which is the orientational contribution in the isotropic case, it was mentioned also in the previous point in our answer.
8. This is an argument based on the behavior of the pressure.
9. Yes, the Referee is right.
10. We apologise for the misleading mistake. Of course the sentence reads correctly as follows: "... for a weakly jammed state the density can increase without increasing the total length of the system previously ...". Thanks for the Referee to point out this crucial mistake. However, due to the modification related to the next point of the Referee, this part was completely deleted from the new version of the manuscript.
11. We accept the Referee's arguments, and delet this part from the manuscript.
12. This part has been rewritten.
13. The mentioned reference is removed.
14. We accept the Referee's suggestion, the "In" is removed.
15. Thanks, it is corrected.
Author: Péter Gurin on 2024-12-04 [id 5025]
(in reply to Report 2 on 2024-11-27)We thank the Referee for the positive report and the remarks with which we agree. We have done the requested changes.

---

## Round 1 · Referee Report · Anonymous (Referee 2) · 2024-11-27

Strengths
2-A good amount of technical details is provided.
3-Monte Carlo simulations nicely validate the transfer operator method calculations.
Weaknesses
2-There are formatting issues in the bibliography.
3-No code/data availability statement is provided.
Report
Requested changes
1-As properly mentioned on p. 9: "The maximum of Pn/P2 is not a sharp transition point between the isotropic and tetratic phase (as expected), but it divides the regions of weakly and strongly ordered structures." In fact, phase transitions are impossible in a one-dimensional system with finite-range interactions. Therefore, for all models considered in this work, a single equilibrium phase is observed. Different regimes certainly exist, but they are separated by a crossover associated with subdominant eigenvalues crossing and splitting, not by an actual phase transition associated with a (high-order) singularity in lambda_0. The text nevertheless refers to"phase changes" throughout, starting from its title. To avoid any confusion, instances of "phase changes" should be replaced by "crossover" and instances of "phase" should be replaced by "regime". A brief clarifying discussion should also be included.
2-In the bibliography, article titles should be properly capitalized. For instance, the title of Ref. [22] should be: "Two-step melting in two...", that of Ref. [35] should be "Monte Carlo study...", and that of Ref. [53] should be "Comment on "Kosterlitz-Thouless-type...".
3-Some references, in addition to the doi, include a link to a pdf, e.g., Refs. [7], [17], and [55]. That second link should be removed.
4-Journal titles should be properly abbreviated, e.g., in Ref. [16], Chemical Physics Letters should be Chem. Phys. Lett.
5-I would expect a paper of this type to make the underlying data and code available in a public repository, but that does not seem to be so. In any case, a data availability statement should be provided.
Recommendation
Ask for minor revision

---

## Round 2 · Author Response

We thank again the Reviewers and the Editor for the helpful suggestions, we feel that the quality of our manuscript has been improved by their work.

---

## Round 2 · List of Changes

According to the requests of the Referees, we have done the following changes.

  1. Eqs. 2a and 2b are corrected.

  2. The de Broglie thermal wavelength is included in Eq. 7, thus the dimensionality is correct.

  3. Our initial guess for the eigenfunction is corrected. (pg. 5)

  4. Fig. 5 is changed, for the details see our response to report #1. According to the new figure, the discussion about d_{max} is removed from the main text.

  5. We include information about the highest pressure simulated, and a sentence about the errors, see the caption of Fig. 3.

  6. To avoid confusion, we defined explicitly what we mean by the different entropy terms (pg. 9 from line 2 to line 10.), according to our response to report #1. Moreover, also in the Introduction we everywhere refer to the "packing vs. orientational entropy" competition, instead of the translational entropy.

  7. We removed the criticized discussion about the strongly and weakly jammed states, together with Fig. 9. of the original manuscript.

  8. On pg. 2 of the Introduction, the second sentence of paragraph 2 has been rephrased to improve context.

  9. Ref. 33 of the original manuscript is removed.

  10. The typos mentioned in report #1 (points 14 and 15) are corrected.

  11. According to the request of report #2 we everywhere replaced the term "phase" by "regime" and the "phase changes" is replaced by "crossover".

  12. The list of references is corrected according to the points 2, 3 and 4 of report #2.

  13. A "Data and code availability" subsection is included at the end of the manuscript.

---

## Editorial Decision

published